# On Interpolative Hardy-Rogers Type Contractions

**Erdal Karapınar [1,2,*]** 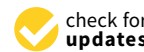**, Obaid Alqahtani [3] and Hassen Aydi [4]**

[1]  Department of Mathematics, Atilim University, Incek, 06830 Ankara, Turkey
[2]  Department of Medical Research, China Medical University Hospital, China Medical University, Taichung 40402, Taiwan
[3]  Department of Mathematics, King Saud University, Riyadh 11451, Saudi Arabia; obalgahtani@ksu.edu.sa
[4]  Department of Mathematics, College of Education in Jubail, Imam Abdulrahman Bin Faisal University, P.O. 12020, Industrial Jubail 31961, Saudi Arabia; hmaydi@iau.edu.sa or hassen.aydi@isima.rnu.tn
*  Correspondence: erdalkarapinar@yahoo.com

**Abstract:** By using an interpolative approach, we recognize the Hardy-Rogers fixed point theorem in the class of metric spaces. The obtained result is supported by some examples. We also give the partial metric case, according to our result.

**Keywords:** metric space; Hardy-Rogers type; fixed point; interpolation; partial metric space

**MSC:** 46T99; 47H10; 54H25

## 1. Introduction and Preliminaries

A Banach couple is two Banach spaces $A$ and $B$ algebraically and topologically imbedded in a separated topological linear space, and denoted by $(A, B)$. The Banach space $E$ is said to be intermediate for the spaces of the Banach couple $(A, B)$ if the imbedding $A \cap B \subset E \subset A + B$ holds.

Let $(A, B)$ and $(C, D)$ be two Banach couples. A linear mapping $T$ acting from the space $A + B$ to $C + D$ is called a *bounded operator from* $(A, B)$ *to* $(C, D)$ if the restrictions of $T$ to the spaces $A$ and $B$ are bounded operators from $A$ to $C$ and $B$ to $D$, respectively.

We denote by $L(AB, CD)$ the linear space of all bounded operators from the couple $(A, B)$ to the couple $(C, D)$. This is a Banach space in the norm

$$\|T\|_{L(AB,CD)} = \max \left\{ \|T\|_{A \to B}, \|T\|_{C \to D} \right\}.$$

**Definition 1** ([1])**.** *Let* $(A, B)$ *and* $(C, D)$ *be two Banach couples, and E (respectively F) be intermediate for the spaces of the Banach couple* $(A, B)$ *(respectively* $(C, D)$*). The triple* $(A, B, E)$ *is called an interpolation triple, relative to* $(C, D, F)$*, if every bounded operator from* $(A, B)$ *to* $(C, D)$ *maps E to F.*

*A triple* $(A, B, E)$ *is said to be an interpolation triple of type* $\alpha$ $(0 \leq \alpha \leq 1)$ *relative to* $(C, D, F)$ *if it is an interpolation triple and the following inequality holds:*

$$\|T\|_{E \to F} \leq c \|T\|_{A \to B}^{\alpha} \cdot \|T\|_{C \to D}^{1-\alpha},$$

*for some constant c.*

Inspired by the definition above, the interpolative Kannan contraction has been described in [2] as follows: Given a metric space $(X, d)$, the mapping $T : X \to X$ is said to be an interpolative Kannan contraction mapping if

$$d(T\theta, T\vartheta) \leq \lambda \left[ d(\theta, T\theta) \right]^{\alpha} \cdot \left[ d(\vartheta, T\vartheta) \right]^{1-\alpha}, \tag{1}$$

for all $\theta, \vartheta \in X$ with $\theta \neq T\theta$, where $\lambda \in [0,1)$ and $\alpha \in (0,1)$. The main result in [2] is the following.

**Theorem 1** ([2]). *Let $(X,d)$ be a complete metric space and $T$ be an interpolative Kannan type contraction. Then $T$ has a unique fixed point in $X$.*

Karapınar, Agarwal and Aydi [3] gave a counter-example to Theorem 1, showing that the fixed point may be not unique. The corrected version of Theorem 1 is the following.

**Theorem 2** ([3]). *Let $(X,d)$ be a complete metric space. Let $T : X \to X$ be a given mapping such that*

$$d\left(T\theta, T\vartheta\right) \leq \lambda \left[d\left(\theta, T\theta\right)\right]^{\alpha} \cdot \left[d\left(\vartheta, T\theta\right)\right]^{1-\alpha},$$

*for all $\theta, \vartheta \in X \backslash Fix(T)$, where $Fix(T) = \{u \in X, Tu = u\}$. Then $T$ has a fixed point in $X$.*

On the other hand, one of generalizations of the Banach Contraction Principle [4] is due to Hardy-Rogers [5].

**Theorem 3.** *Let $(X,d)$ be a complete metric space. Let $T : X \to X$ be a given mapping such that*

$$d\left(T\theta, T\vartheta\right) \leq \alpha d(\theta, y) + \beta d(\theta, T\theta) + \gamma d(y, T\vartheta) + \delta[\frac{1}{2}(d\left(\theta, T\vartheta\right) + d\left(\vartheta, T\theta\right))],$$

*for all $\theta, \vartheta \in X$, where $\alpha, \beta, \gamma, \delta$ are non-negative reals such that $\alpha + \beta + \gamma + \delta < 1$. Then $T$ has a unique fixed point in $X$.*

In this paper, we introduce the concept of interpolative Hardy-Rogers type contractions, and provide some examples illustrating the obtained result. We also extend our obtained result to partial metric spaces.

## 2. Main Results

We start this section by introducing the notion of *interpolative Hardy-Rogers type contractions*.

**Definition 2.** *Let $(X,d)$ be a metric space. We say that the self-mapping $T : X \to X$ is an* interpolative Hardy-Rogers type contraction *if there exists $\lambda \in [0,1)$ and $\alpha, \beta, \gamma \in (0,1)$ with $\alpha + \beta + \gamma < 1$, such that*

$$d\left(T\theta, T\vartheta\right) \leq \lambda \left[d\left(\theta, y\right)\right]^{\beta} \cdot \left[d\left(\theta, T\theta\right)\right]^{\alpha} \cdot \left[d\left(\vartheta, T\theta\right)\right]^{\gamma} \cdot \left[\frac{1}{2}(d\left(\theta, T\vartheta\right) + d\left(\vartheta, T\theta\right))\right]^{1-\alpha-\beta-\gamma} \tag{2}$$

*for all $\theta, \vartheta \in X \backslash Fix(T)$.*

**Theorem 4.** *Let $(X,d)$ be a complete metric space and $T$ be an interpolative Hardy-Rogers type contraction. Then, $T$ has a fixed point in $X$.*

**Proof.** Starting from $\theta_0 \in X$, consider $\{\theta_n\}$, given as $\theta_n = T^n(\theta_0)$ for each positive integer $n$. If there exists $n_0$ such that $\theta_{n_0} = \theta_{n_0+1}$, then $\theta_{n_0}$ is a fixed point of $T$. The proof is completed. So, assume that $\theta_n \neq \theta_{n+1}$ for all $n \geq 0$. By substituting the values $\theta = \theta_n$ and $\vartheta = \theta_{n-1}$ in (2), we find that

$$
\begin{aligned}
d\left(\theta_{n+1}, \theta_n\right) = d\left(T\theta_n, T\theta_{n-1}\right) \quad &\leq \lambda \left[d\left(\theta_n, \theta_{n-1}\right)\right]^{\beta} \left[d\left(\theta_n, T\theta_n\right)\right]^{\alpha} \cdot \left[d\left(\theta_{n-1}, T\theta_{n-1}\right)\right]^{\gamma} \\
&\quad \cdot \left[\frac{1}{2}(d\left(\theta_n, \theta_n\right) + d\left(\theta_{n-1}, \theta_{n+1}\right))\right]^{1-\alpha-\beta-\gamma} \\
\\
&\leq \lambda \left[d\left(\theta_n, \theta_{n-1}\right)\right]^{\beta} \cdot \left[d\left(\theta_n, \theta_{n+1}\right)\right]^{\alpha} \cdot \left[d\left(\theta_{n-1}, \theta_n\right)\right]^{\gamma} \\
&\quad \cdot \left[\frac{1}{2}(d\left(\theta_{n-1}, \theta_n\right) + d\left(\theta_n, \theta_{n+1}\right))\right]^{1-\alpha-\beta-\gamma}.
\end{aligned}
\tag{3}
$$

Suppose that $d(\theta_{n-1}, \theta_n) < d(\theta_n, \theta_{n+1})$ for some $n \geq 1$. Thus,

$$\frac{1}{2}(d(\theta_{n-1}, \theta_n) + d(\theta_n, \theta_{n+1})) \leq d(\theta_n, \theta_{n+1}).$$

Consequently, the inequality (13) yields that

$$[d(\theta_n, \theta_{n+1})]^{\beta+\gamma} \leq \lambda [d(\theta_{n-1}, \theta_n)]^{\beta+\gamma}. \tag{4}$$

So, we conclude that $d(\theta_{n-1}, \theta_n) \geq d(\theta_n, \theta_{n+1})$, which is a contradiction. Thus, we have $d(\theta_n, \theta_{n+1}) \leq d(\theta_{n-1}, \theta_n)$ for all $n \geq 1$. Hence, $\{d(\theta_{n-1}, \theta_n)\}$ is a non-increasing sequence with positive terms. Set $\ell =: \lim_{n \to \infty} d(\theta_{n-1}, \theta_n)$. We have

$$\frac{1}{2}(d(\theta_{n-1}, \theta_n) + d(\theta_n, \theta_{n+1})) \leq d(\theta_{n-1}, \theta_n), \quad \text{for all } n \geq 1.$$

By a simple elimination, the inequality (13) implies that

$$[d(\theta_n, \theta_{n+1})]^{1-\alpha} \leq \lambda [d(\theta_{n-1}, \theta_n)]^{1-\alpha}, \quad \text{for all } n \geq 1. \tag{5}$$

We deduce that

$$d(\theta_n, \theta_{n+1}) \leq \lambda d(\theta_{n-1}, \theta_n) \leq \lambda^n d(\theta_0, \theta_1). \tag{6}$$

On account of the assumption that $\lambda < 1$, by taking $n \to \infty$ in the inequality (15), we get that $\ell = 0$. In what follows, we shall prove that $\{\theta_n\}$ is a Cauchy sequence by employing standard tools. More precisely, starting with the triangle inequality, we shall get the following estimation:

$$\begin{aligned} d(\theta_n, \theta_{n+r}) \ &\leq d(\theta_n, \theta_{n+1}) + \cdots + d(\theta_{n+r-1}, \theta_{n+r}) \\ &\leq \lambda^n d(\theta_0, \theta_1) + \cdots + \lambda^{n+r-1} d(\theta_0, \theta_1) \\ &\leq \frac{\lambda^n}{1-\lambda} d(\theta_0, \theta_1). \end{aligned} \tag{7}$$

Thus, $\{\theta_n\}$ is a Cauchy sequence in the complete metric space $(X, d)$, and so there exists $\theta \in X$ such that $\lim_{n \to \infty} d(\theta_n, \theta) = 0$. Suppose that $\theta \neq T\theta$. Since $\theta_n \neq T\theta_n$ for each $n \geq 0$, by letting $\theta = \theta_n$ and $\vartheta = \theta$ in (2), we have

$$\begin{aligned} d(\theta_{n+1}, T\theta) = d(T\theta_n, T\theta) &\leq \lambda [d(\theta_n, \theta)]^{\beta} \cdot [d(\theta_n, T\theta_n)]^{\alpha} \cdot [d(\theta, T\theta)]^{\gamma} \\ &\cdot \left[\frac{1}{2}(d(\theta_{n+1}, T\theta) + d(\theta, T\theta_{n+1}))\right]^{1-\alpha-\beta-\gamma}. \end{aligned} \tag{8}$$

Letting $n \to \infty$ in the inequality (19), we find that $d(\theta, T\theta) = 0$, which is a contradiction. Thus, $T\theta = \theta$. $\square$

In what follows, we shall consider the analog of Theorem 4, in the setting of partial metric spaces. For this purpose, we recall the fundamental notions and basic observations.

**Definition 3** (See [6]). *Let X be a non-empty set. A function $p : X \times X \to [0, \infty)$ is said to be a partial metric if the following conditions are fulfilled, for each $\xi, \eta, \zeta \in X$:*

$$\begin{aligned} &(P1) \quad \xi = \eta \Leftrightarrow p(\xi, \xi) = p(\eta, \eta) = p(\xi, \eta); \\ &(P2) \quad p(\xi, \xi) \leq p(\xi, \eta); \\ &(P3) \quad p(\xi, \eta) = p(\eta, \xi); \\ &(P4) \quad p(\xi, \eta) \leq p(\xi, \zeta) + p(\zeta, \eta) - p(\zeta, \zeta). \end{aligned} \tag{9}$$

In this case, $(X, p)$ is said to be a partial metric space.

The function $\rho_p : X \times X \to [0, \infty)$, defined as

$$\rho_p(\xi, \eta) = 2p(\xi, \eta) - p(\xi, \xi) - p(\eta, \eta) \tag{10}$$

is a standard metric on $X$. It is natural to define the basic topological concepts, in particular, convergence of a sequence, fundamental (Cauchy) sequence criteria, continuity of the mappings, and completeness of the topological space, in the framework of partial metric spaces; see, for example, [7–16].

**Definition 4.** *In the framework of a partial metric space $(X, p)$, we say that*

(i)   *a sequence $\{\xi_n\}$ converges to the limit $\xi$ if $p(\xi, \xi) = \lim\limits_{n \to \infty} p(\xi, \xi_n)$;*

(ii)   *a sequence $\{\xi_n\}$ is fundamental (or Cauchy) if $\lim\limits_{n,m \to \infty} p(\xi_n, \xi_m)$ exists and is finite;*

(iii)   *a partial metric space $(X, p)$ is complete if each fundamental sequence $\{\xi_n\}$ converges to a point $\xi \in X$ such that $p(\xi, \xi) = \lim\limits_{n,m \to \infty} p(\xi_n, \xi_m)$; and*

(iv)   *a mapping $F : X \to X$ is continuous at a point $\xi_0 \in X$ if, for each $\epsilon > 0$, there exists $\delta > 0$ such that $F(B_p(\xi_0, \delta)) \subseteq B_P(F\xi_0, \epsilon)$.*

In what follows, we shall recall the following easily-derived lemma (see [6]).

**Lemma 1.** *Let $p$ be a partial metric on a non-empty set $X$ and $\rho_p$ be the corresponding standard metric space on the same set $X$.*

(a)   *A sequence $\{\xi_n\}$ is fundamental in the framework of a partial metric $(X, p)$ if and only if it is a fundamental sequence in the setting of the corresponding standard metric space $(X, \rho_p)$.*

(b)   *A partial metric space $(X, p)$ is complete if and only if the corresponding standard metric space $(X, \rho_p)$ is complete. Moreover,*

$$\lim\limits_{n \to \infty} \rho_p(\xi, \xi_n) = 0 \Leftrightarrow p(\xi, \xi) = \lim\limits_{n \to \infty} p(\xi, \xi_n) = \lim\limits_{n,m \to \infty} p(\xi_n, \xi_m). \tag{11}$$

(c)   *If $\xi_n \to \zeta$ as $n \to \infty$ in a partial metric space $(X, p)$ with $p(\zeta, \zeta) = 0$, then we have*

$$\lim\limits_{n \to \infty} p(\xi_n, \eta) = p(\zeta, \eta) \text{ for every } \eta \in X.$$

The following theorem is an analog of Theorem 4, in the setting of partial metric spaces.

**Theorem 5.** *Let $(X, p)$ be a completed partial metric space. Let $T : X \to X$ be a given mapping. Suppose there exists $\lambda \in [0, 1)$ and $\alpha, \beta, \gamma \in (0, 1)$ with $\alpha + \beta + \gamma < 1$, such that*

$$p\,(T\xi, T\eta) \leq \lambda \,[p\,(\xi, \eta)]^\beta \cdot [p\,(\xi, T\xi)]^\alpha \cdot [p\,(\eta, T\eta)]^\gamma \cdot \left[ \frac{1}{2}(p\,(\xi, T\eta) + p\,(\eta, T\xi)) \right]^{1-\alpha-\beta-\gamma}, \tag{12}$$

*for all $\xi, \eta \in X \backslash Fix(T)$. Then, $T$ has a fixed point in $X$.*

**Proof.** For any $\xi_0 \in (X, p)$, we construct a sequence $\{\xi_n\}$ by $\xi_n = T^n(\xi_0)$ for each $n \in \mathbb{N}$. If there exists $n_0$ such that $\xi_{n_0} = \xi_{n_0+1}$, then $\xi_{n_0}$ is a fixed point of $T$. The proof is completed. So, assume that $\xi_n \neq \xi_{n+1}$ for each $n \geq 0$. By substituting the values $\xi = \xi_n$ and $\eta = \xi_{n-1}$ in (12), we find that

$$
\begin{aligned}
p\left(\xi_{n+1}, \xi_{n}\right)=p\left(T \xi_{n}, T \xi_{n-1}\right) \quad & \leq \lambda\left[p\left(\xi_{n}, \xi_{n-1}\right)\right]^{\beta} \cdot\left[p\left(\xi_{n}, T \xi_{n}\right)\right]^{\alpha} \cdot\left[p\left(\xi_{n-1}, T \xi_{n-1}\right)\right]^{\gamma} \\
& \cdot\left[\tfrac{1}{2}\left(p\left(\xi_{n}, T \xi_{n-1}\right)+p\left(\xi_{n-1}, T \xi_{n}\right)\right)\right]^{1-\alpha-\beta-\gamma} \\[4pt]
& =\lambda\left[p\left(\xi_{n}, \xi_{n-1}\right)\right]^{\beta} \cdot\left[p\left(\xi_{n}, \xi_{n+1}\right)\right]^{\alpha} \cdot\left[p\left(\xi_{n-1}, \xi_{n}\right)\right]^{\gamma} \\
& \cdot\left[\tfrac{1}{2}\left(p\left(\xi_{n}, \xi_{n-1}\right)+p\left(\xi_{n+1}, \xi_{n}\right)\right)\right]^{1-\alpha-\beta-\gamma} .
\end{aligned}
\tag{13}
$$

Now, if we suppose that $p\left(\xi_{n-1}, \xi_{n}\right) \leq p\left(\xi_{n}, \xi_{n+1}\right)$, then the inequality (13) yields $p\left(\xi_{n}, \xi_{n+1}\right) \leq \lambda p\left(\xi_{n}, \xi_{n+1}\right)$, a contradiction (since $\lambda<1$). Thus, we conclude $p\left(\xi_{n}, \xi_{n+1}\right) \leq p\left(\xi_{n-1}, \xi_{n}\right)$, that is, $\left\{p\left(\xi_{n-1}, \xi_{n}\right)\right\}$ is a non-increasing sequence. Accordingly, we get, by inequality (13), that

$$
\left[p\left(\xi_{n}, \xi_{n+1}\right)\right]^{1-\alpha} \leq \lambda\left[p\left(\xi_{n-1}, \xi_{n}\right)\right]^{1-\alpha}
\tag{14}
$$

Thus, there is a nonnegative constant $\ell$ such that $\lim_{n \rightarrow \infty} p\left(\xi_{n-1}, \xi_{n}\right)=\ell$. Note that $\ell \geq 0$. Notice that (14) yields

$$
p\left(\xi_{n}, \xi_{n+1}\right) \leq \lambda^{\alpha} p\left(\xi_{n-1}, \xi_{n}\right) \leq \lambda^{n \alpha} p\left(\xi_{0}, \xi_{1}\right) .
\tag{15}
$$

Since $\lambda, \alpha<1$ we have $\tilde{\lambda}:=\lambda^{\alpha}<1$. Hence, by letting $n \rightarrow \infty$ in (15), we get that $\ell=0$.

We shall use the modified triangle inequality of the partial metric to prove that $\left\{\xi_{n}\right\}$ is a fundamental (Cauchy) sequence:

$$
\begin{aligned}
p\left(\xi_{n}, \xi_{n+r}\right) \quad & \leq p\left(\xi_{n}, \xi_{n+1}\right)+\cdots+p\left(\xi_{n+r-1}, \xi_{n+r}\right) \\[4pt]
& \leq \tilde{\lambda}^{n} p\left(\xi_{0}, \xi_{1}\right)+\cdots+\tilde{\lambda}^{n+r-1} p\left(\xi_{0}, \xi_{1}\right) \\[4pt]
& \leq \frac{\tilde{\lambda}^{n}}{1-\tilde{\lambda}} p\left(\xi_{0}, \xi_{1}\right) .
\end{aligned}
\tag{16}
$$

We conclude that $\left\{\xi_{n}\right\}$ is a fundamental sequence in $(X, p)$, by taking $n \rightarrow \infty$. By Lemma 1, $\left\{\xi_{n}\right\}$ is fundamental sequence in the corresponding standard metric $\left(X, \rho_{p}\right)$. More particularly, since $(X, p)$ is complete, $\left(X, \rho_{p}\right)$ is also complete. Hence, there exists $\xi \in X$ such that

$$
p(\xi, \xi)=\lim_{n \rightarrow \infty} p\left(\xi, \xi_{n}\right)=\lim_{n, m \rightarrow \infty} p\left(\xi_{n}, \xi_{m}\right)=0,
\tag{17}
$$

which implies that

$$
\lim_{n \rightarrow \infty} \rho_{p}\left(\xi, \xi_{n}\right)=0 .
\tag{18}
$$

As a next step, we make evident that the limit $\xi$ of the iterative sequence $\left\{\xi_{n}\right\}$ is a fixed point of the given mapping $T$. Assume that $\xi \neq T \xi$, so $p(\xi, T \xi)>0$. Recall that $\xi_{n} \neq T \xi_{n}$ for each $n \geq 0$. By letting $\xi=\xi_{n}$ and $\eta=\xi$ in (12), we determine that

$$
\begin{aligned}
p\left(\xi_{n+1}, T \xi\right)= & p\left(T \xi_{n}, T \xi\right) \\
\leq & \lambda\left[p\left(\xi_{n}, \xi\right)\right]^{\beta} \cdot\left[p\left(\xi_{n}, T \xi_{n}\right)\right]^{\alpha} \cdot\left[p(\xi, T \xi)\right]^{\gamma} \\
& \cdot\left[\frac{1}{2}\left(p\left(\xi_{n}, T \xi\right)+p\left(\xi, T \xi_{n}\right)\right)\right]^{1-\alpha-\beta-\gamma} \\
= & \lambda\left[p\left(\xi_{n}, \xi\right)\right]^{\beta} \cdot\left[p\left(\xi_{n}, \xi_{n+1}\right)\right]^{\alpha} \cdot\left[p(\xi, T \xi)\right]^{\gamma} \\
& \cdot\left[\frac{1}{2}\left(p\left(\xi_{n}, T \xi\right)+p\left(\xi, \xi_{n+1}\right)\right)\right]^{1-\alpha-\beta-\gamma} .
\end{aligned}
\tag{19}
$$

Letting $n \rightarrow \infty$ in the inequality (19), we find that $p(\xi, T \xi)=0$, and so $\xi=T \xi$, which is a contradiction. Thus, $T \xi=\xi$. $\square$

In the following examples, the fixed point exists but is not unique.

**Example 1.** *Consider $X = \{0, 1, 2, 3, 5\}$ endowed with $d(\theta, \vartheta) = |\theta - \vartheta|$. Choose $\lambda = \frac{\sqrt{2}}{2}$, $\alpha = \frac{1}{3}$, $\beta = \frac{1}{2}$ and $\gamma = \frac{1}{7}$. It is obvious that*

$$d(T\theta, T\vartheta) \leq \lambda \left[d(\theta, y)\right]^{\beta} \cdot \left[d(\theta, T\theta)\right]^{\alpha} \cdot \left[d(\vartheta, T\vartheta)\right]^{\gamma} \cdot \left[\frac{1}{2}(d(\theta, T\vartheta) + d(\vartheta, T\theta))\right]^{1-\alpha-\beta-\gamma},$$

*for all $\theta, \vartheta \in X \backslash Fix(T)$; that is, (2) holds. All the hypotheses of Theorem 4 are satisfied, and so T has a fixed point. Here, we have two fixed points, which are 0 and 1.*

*On the other hand, for $\theta = 0$ and $\vartheta = 1$, we have*

$$d(T\theta, T\vartheta) > \lambda \left[d(\theta, \vartheta) + d(\theta, T\theta) + d(\vartheta, T\vartheta) + \frac{1}{2}(d(\theta, T\vartheta) + d(\vartheta, T\theta))\right],$$

*for any $\lambda \in [0, \frac{1}{4})$, so Theorem 3 (for $\lambda = \alpha = \beta = \gamma = \delta$) is not applicable.*

**Example 2.** *Let $X = [0, \infty)$ be endowed with the metric*

$$d(\theta, y) = \begin{cases} 0 & if \quad \theta = \vartheta \\ 1 & if \quad \theta \neq \vartheta. \end{cases}$$

*Define the self-mapping on X as*

$$T\theta = \begin{cases} 0 & if \quad \theta \in [0, 1) \\ \theta & if \quad \theta \geq 1. \end{cases}$$

*Let $\theta, \vartheta \in X \backslash Fix(T)$. Then $\theta, \vartheta \notin (0, 1)$, and so $d(T\theta, T\vartheta) = 0$; that is, (2) holds. Thus, all the hypotheses of Theorem 4 hold, and so T has a fixed point. Here, we have an infinite number of fixed points.*

*On the other hand, Theorem 3 is not applicable (it suffices to take $\theta = 0$ and $\vartheta = 1$).*

**Remark 1.** *It is known that Theorem 3 is a generalization of the Banach, Kannan, and Reich fixed point results. Note that, in our new approach via interpolation, there is no relation between all of the above results using the interpolative approach. That is, Theorem 4 is totally independent of Corollary 2.1 in [3] and Theorem 2.*

## 3. Conclusions

We aim to enrich the fixed point theory by involving interpolative approaches.

**Author Contributions:** All authors contributed equally and significantly in writing this article. All authors read and approved the final manuscript.

**Funding:** This research received no external funding.

**Acknowledgments:** The authors thanks to anonymous referees for their remarkable comments, suggestion and ideas that helps to improve this paper. The authors extend their appreciation to the Deanship of Scientific Research at King Saud University for funding this work through research group No RG-1440-025.

**Conflicts of Interest:** The authors declare no conflict of interest.

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
