# Peer review of "On Interpolative Hardy-Rogers Type Contractions"

_symmetry, doi:10.3390/sym11010008_

Reviewer 1 Report

In the first line of Theorem 5 , (X,d) must be changed as (X,p)

The proof of Theorem 5 should be provided.

The motivation for Theorems 4 and 5 are not clear. The authors need to explain why we need this new type of fixed point theorem. Maybe the authors can provide some applications in nonlinear analysis.

Author Response

Dear Editor,

We are thankful to the anonymous reviewers for their careful reading and remarks on our manuscript.

We revised the manuscript with respect to the suggestions and remarks of the reviewer.  

We corrected (X,d) as (X,p) in Theorem 5 and we put the proof of Theorem 5.

We put some lines into the introduction section to express our motivation of usage of interpolation approaches.

Reviewer 2 Report

The following should be corrected:

(a) re-write the authors' full addresses.

(b) p.1(+6), 31961, Saudi Arabia

(4) type, fixed

(6) is said to be an

(7) is the following.

(10) Karapinar, Agarwal and Aydi [4] gave a

(11) is the following.

(24) `Letting -- in the inequality (7) -- ' should be replaced by `Thus $\{\theta_n\}$ is a Cauchy sequence --'

(27) is the following.

(30) All the

(33) all the

(47) point theorem

(49) -- type contractions via interpolation,

(52), spaces (preprint).

Author Response

a)    We re-wrote the addresses

b)     we corrected “dot” with “comma

(4) “Fixed point” correct as “fixed point”

(6)   “said an interpolative”  corrected as   “said to be an interpolative

(7)  “is.”   is corrected as “is the following.”

(10) the authors in [ 4 ] is corrected as Karapinar, Agarwal and Aydi [4

(11)  is.”   is corrected as  “is the following”.

(24) `We corrected “Letting -- in the inequality (7) -- ' should as  `Thus $\{\theta_n\}$ is a Cauchy sequence --'

(27)  is.”   is corrected as “is the following.”

(30) “All “ is replaced with  “All the”

(33)   ”all” is repleced with  “all the”

(47)  “fixed point Theorem”corrected as  “fixed point theorem”

(49) -We modifed “Type Contractions via Interpolation” as - type contractions via interpolation,

(52), spaces (preprint). we put the journal name and volumes.

Reviewer 3 Report

The main result of this paper is new, correct and significant. This paper surely deserves publication in its current form.

Author Response

Thank you so much.

Reviewer 4 Report

The Authors suggested the interpolation approach to one of the known contractions. The project has some merit, but I think that there is needed more motivation for introducing another type of contractions. Please, justify by providing some kind of application or explain in details why the rate of convergence has been improved.

Author Response

Reviwer: The Authors suggested the interpolation approach to one of the known contractions. The project has some merit, but I think that there is needed more motivation for introducing another type of contractions. 

Authors: Roughly speaking the reviewer is right. Mainly, We aim to  enrich the fixed point theory by inolving the approaches/ techniques of the interpolation theory  to the fixed point theory. Interpolation theory can help us to get a better estimation in calculation. This is our first motivation to consider this paper.

Reviewer: Please, justify by providing some kind of application or explain in details why the rate of convergence has been improved.

Authors: Concrete calculation is our mind but we thought it to next step.Indeed, interpolation theory is very deep. We are mainly inspired from the renowned book:

Krein, S.G, Petunin, J.I, Semenov, E.M .: Interpolation Of Linear Operators, American Mathematical Society, 103 Providence, Rhode Island (1978).

We aim to go through but, as the reviewers appreciate, it takes time.

Round  2

Reviewer 1 Report

The reviewer is satisfied with the revision.

Reviewer 4 Report

I accept the paper in the present form and recommend the acceptance.